## Research Article

psychosocial; implementation research; implementation strategy; displacement; migration

**Corresponding author:**
Claire Greene;
Email: Mg4069@cumc.columbia.edu

# Strategies to improve the implementation and effectiveness of community-based psychosocial support interventions for displaced, migrant and host community women in Latin America

M. Claire Greene[1] , Gabrielle Wimer[2], Maria Larrea[3], Ingrid Mejia Jimenez[1], Andrea Armijos[4], Alejandra Angulo[5], Maria Esther Guevara[4], Carolina Vega[4,6], Emily W. Heard[1], Lina Demis[1], Lucia Benavides[4], Christine Corrales[5], Ale de la Cruz[4], Maria Jose Lopez[4], Arianna Moyano[4], Andrea Murcia[5], Maria Jose Noboa[4], Abhimeleck Rodriguez[5], Jennifer Solis[5], Daniela Vergara[4], Lena S. Andersen[6] , Maria Cristobal[7], Milton Wainberg[8], Annie G. Bonz[7] and Wietse Tol[6,9]

[1]Program on Forced Migration and Health, Heilbrunn Department of Population and Family Health, Columbia University Mailman School of Public Health, New York, NY, USA; [2]Columbia University Vagelos College of Physicians and Surgeons, New York, NY, USA; [3]Hunter College, City University of New York, New York, NY, USA; [4]HIAS Ecuador, Quito, Ecuador; [5]HIAS Panamá, Panamá City, Panamá; [6]Global Health Section, Department of Public Health, University of Copenhagen, Copenhagen, Denmark; [7]HIAS, Silver Spring, MD, USA; [8]Department of Psychiatry, Columbia University Vagelos College of Physicians and Surgeons/New York State Psychiatric Institute, New York, NY, USA and [9]Athena Research Institute, Vrije Universiteit Amsterdam, Amsterdam, The Netherlands

## Abstract

As evidence supporting the effectiveness of mental health and psychosocial interventions grows, more research is needed to understand optimal strategies for improving their implementation in diverse contexts. We conducted a qualitative process evaluation of a multicomponent psychosocial intervention intended to promote well-being among refugee, migrant and host community women in three diverse contexts in Ecuador and Panamá. The objective of this study is to describe the relationships among implementation determinants, strategies and outcomes of this community-based psychosocial intervention. The five implementation strategies used in this study included stakeholder engagement, promoting intervention adaptability, group and community-based delivery format, task sharing and providing incentives. We identified 10 adaptations to the intervention and its implementation, most of which were made during pre-implementation. Participants (n = 77) and facilitators (n = 30) who completed qualitative interviews reported that these strategies largely improved the implementation of the intervention across key outcomes and aligned with the study's intervention and implementation theory of change models. Participants and facilitators also proposed additional strategies for improving reach, implementation and maintenance of this community-based psychosocial intervention.

## Impact statement

Evidence supporting the effectiveness of psychosocial interventions for improving mental health and well-being in humanitarian emergencies is growing. Barriers to implementing and sustaining these interventions remain that may compromise the adoption and sustainability of these interventions within routine, community-based services. Further research on promising strategies for improving the implementation of these interventions is needed. This study aims to describe the implementation of a community-based psychosocial intervention and to examine a set of implementation strategies that were used to deliver this intervention to 225 displaced, refugee, migrant and host community women in Ecuador and Panama. The strategies included the following: (1) engaging community stakeholders throughout intervention design and implementation; (2) promoting intervention adaptability, usability and fit; (3) delivering the intervention within community settings and in a group format; (4) training community members to deliver the intervention using a task-sharing model with ongoing training and supervision and 5) providing compensation to the intervention facilitators for their time and reimbursing participant costs associated with participation in the intervention (e.g., transportation and communication vouchers). We conducted qualitative interviews with women who participated in this intervention (n = 77) and women who delivered the intervention (i.e., facilitators, n = 30). Using the RE-AIM and PRISM frameworks, we coded and connected the characteristics of the implementation determinants, strategies and outcomes. Women

reported that this set of five strategies improved various aspects of the intervention's implementation, including its reach, effectiveness, adoption, implementation and maintenance. Many of the perceived impacts of these strategies were similar across sites. However, certain aspects of each of the study contexts (e.g., population mobility, community insecurity) necessitated specific adaptations or resulted in differences in implementation outcomes. Further mixed-methods research comparing implementation strategies is needed to advance the evidence on how to best deliver community-based psychosocial interventions in diverse humanitarian contexts.

## Introduction

Regional migration within Latin America has considerably increased in recent years (International Organization for Migration, 2023). As of March 2023, over seven million migrants, refugees and asylum seekers had fled the economic and political crisis in Venezuela, with approximately six million remaining within Latin America and the Caribbean (R4V, 2023). In Central America and the Caribbean, violence, insecurity, poverty and the increased risk of climate-related emergencies have contributed to displacement within the region (Bojorquez et al., 2021; acaps, 2022). The population of migrants in the America and their reasons for migration are diverse (International Organization for Migration, 2023). Ecuador and Panamá are among the largest host and transit countries in the region. Ecuador is host to over 500,000 refugees and migrants in need of international protection, primarily from Colombia and Venezuela (International Organization for Migration, 2023; United Nations High Commissioner for Refugees, 2023). Migration through the Darién region of Panamá has increased in recent years. In 2023, an estimated 1,572 migrants cross from Colombia into this region of Panamá daily, most of whom are from Venezuela, followed by Ecuador, Haiti, China and Colombia (República de Panamá, 2023).

Populations affected and displaced by humanitarian emergencies experience an elevated risk of mental health problems (Kirmayer et al., 2011; Charlson et al., 2019; Mesa-Vieira et al., 2022). Recent quantitative and qualitative research on mental health among migrants and refugees in Latin America, including migrants in Ecuador and Panamá, has identified symptoms of depression (Carroll et al., 2020; Greene et al., 2022b), generalized anxiety disorder (Carroll et al., 2020), post-traumatic stress disorder (Espinel et al., 2020) and general psychological distress, including feelings of fear, anger and stress (Greene et al., 2022b; Mougenot et al., 2021). Migrants and forcibly displaced populations in Latin America face a range of risk factors for mental health problems across the phases of migration, including exposure to potentially traumatic events, discrimination and xenophobia, social isolation, lack of integration and socioeconomic adversity (Keller et al., 2017; Carroll et al., 2020; Mougenot et al., 2021; Morales et al., 2022; Salas-Wright et al., 2022). There is growing evidence supporting the effectiveness of community-based mental health and psychosocial interventions for alleviating symptoms of common mental disorders and psychological distress among displaced populations (Bangpan et al., 2019; Turrini et al., 2019; Barbui et al., 2020; Haroz et al., 2020), including for displaced and emergency-affected populations in Latin America (Bonilla-Escobar et al., 2018; 2023). However, access to and utilization of mental health and psychosocial support services to prevent and treat mental health problems remains limited (Cubides et al., 2022). Barriers include lack of information about how and where to seek services, inability to access care due to legal or migratory status, lack of mental health and psychosocial support providers, disparities in insurance coverage and insufficient resources or capacity to address the needs of displaced persons (e.g., refugees, asylum seekers) and migrants within host country health systems (Kohrt et al., 2020; Agudelo-Suarez et al., 2022; Blukacz et al., 2022; Bowser et al., 2022;

Cubides et al., 2022). Even in host countries that have instituted pathways for migrants and displaced populations to access services provided within national health systems, disparities in health insurance coverage and utilization of health services persist (Bowser et al., 2022; Shepard et al., 2023).

Innovative strategies are needed to overcome these barriers and improve access to and implementation of mental health and psychosocial support programs for migrants and displaced persons in humanitarian settings. Many of the strategies tested in humanitarian mental health research have focused on building the capacity of nonspecialist health providers or other personnel to deliver mental health interventions to address the often-limited number of mental health providers within health systems and community settings (Cohen and Yaeger, 2021). Other strategies that have been documented in mental health and psychosocial support research in humanitarian contexts include a range of stakeholder engagement strategies (Dickson and Bangpan, 2018), methodologies for adapting interventions across populations and contexts (Sangraula et al., 2021), along with other capacity building strategies (Echeverri et al., 2018). Due to the unique challenges of providing mental health and psychosocial support to mobile populations and in emergency contexts, the relationship between the strategies used and implementation outcomes is rarely empirically explored.

The objective of this study is to describe the relationship among implementation determinants, strategies and outcomes of a community-based psychosocial intervention for displaced, migrant and host community women in Ecuador and Panamá.

## Methods

This process evaluation examines the implementation of two variations of a group psychosocial intervention, *Entre Nosotras* ('among/between us'), which was developed for refugee, migrant and host community women in Latin America. Variations of this intervention were comparatively evaluated using a cluster randomized feasibility trial design across 11 communities nested within 3 sites in Ecuador and Panamá (Greene et al., 2022a). The three sites were selected to represent variation in host communities by displacement dynamics, urbanization, available mental health and psychosocial services and population characteristics. The communities within Panamá included the capital city and three peri-urban communities surrounding the capital, where many refugees and migrants have settled. In Ecuador, the study sites included urban communities in Guayaquil, a destination city for many refugees and migrants and rural communities in Tulcán, which is often a temporary place of transit for refugees and migrants. All study activities were designed and implemented in partnership with HIAS, an international humanitarian organization that provides community-based mental health and psychosocial support services in Ecuador and Panamá.

The intervention, *Entre Nosotras*, is a five-session group intervention co-designed with refugee, migrant and host community members from the study communities to improve mental health, sense of safety, community connectedness and social support. The intervention combined elements of psychoeducation, stress

**Box 1.** Description of implementation strategies (Table 1).

*Strategy 1. Engage community stakeholders throughout intervention design and implementation.*

Community leaders and members as well as local, national, regional and global implementation staff and technical advisors were continuously engaged throughout the phases of implementation. Prior to implementation, we conducted free listing interviews with 97 community members to align the intervention targets to local priorities. We conducted 36 in-depth interviews with community leaders, mental health providers, representatives of community-based organizations, government officials, police and other stakeholders to explore relevant responses to address priority mental health and protection issues in these communities (Greene et al., 2022b). We conducted participatory intervention design workshops (see Strategy #2) with members of the community to develop the intervention. The study team maintained regular communication with study participants and facilitators, community leaders, HIAS staff and external stakeholders throughout the implementation and regularly requested feedback and recommendations. At the end of the study period, we conducted in-depth interviews with 107 study participants and facilitators to gather their input and conducted community dissemination events to share and discuss preliminary findings from the study with community, academic and humanitarian audiences.

*Strategy 2. Iterative co-design of the intervention.*

We assembled groups of 10–15 migrant women within each of the study communities to participate in iterative intervention co-design workshops that involved brainstorming intervention targets and mechanisms, developing a theory of change, identifying and tailoring intervention components, pilot testing these intervention components through mock sessions and refining intervention elements. This strategy enabled each of the sites to tailor elements of the intervention to their context, which was introduced to promote adaptability and usability. We noted the similarities and differences that emerged across study sites and incorporated recommendations within the manual for elements that could be modified to fit different contextual realities (e.g., population differences, in person vs. hybrid sessions, etc.). Throughout the intervention implementation period, we documented adaptations that were made and requested feedback and recommendations from participants and facilitators to improve the intervention. Details of the intervention co-design process are published elsewhere (Greene et al., 2022b).

*Strategy 3. Deliver the intervention within community settings and in a group format.*

Based on the results from the formative research (including Strategies #1 and 2), we identified secure spaces within each study community that would make the intervention more accessible. We also designed the intervention to be delivered in groups by a pair of facilitators to align with the preferences of the target population.

*Strategy 4. Train community members to deliver the intervention using a task-sharing model with ongoing training and supervision.*

Thirty-two refugee, migrant and host community women were invited to be trained as facilitators by HIAS staff. They were identified through a variety of mechanisms including recommendations from community leaders, participation in the formative research and intervention co-design (see Strategies #1 and 2) or were involved in community-based organizations. HIAS program staff, including at least one psychologist in each site, conducted an initial 2-week training with facilitators in person in each of the sites. The initial training involved didactic learning and role plays. Facilitator competencies were assessed at the end of the training to ensure sufficient mastery of the intervention content and basic counseling skills (Kohrt et al., 2015). The facilitators received weekly group supervision throughout the implementation period and were in regular contact with the study staff. A detailed description of the training and supervision as well as the results of the feasibility trial are published elsewhere (Greene et al., 2022b; 2023).

*Strategy 5. Provide compensation to the intervention facilitators for their time and reimburse participant costs associated with participation in the intervention (e.g., transportation and communication vouchers).*

All facilitators received a stipend for their time and effort in delivering the Entre Nosotras intervention. Participants were not compensated for their participation, but instead received vouchers or reimbursements for transportation or communication costs they incurred to participate in the study activities.

management, individual and community problem solving and other participatory methodologies focused on promoting well-being (Figure 1a) (Greene et al., 2022b). Implementation of the intervention involved five strategies that were designed based on formative research and intended to improve the reach, effectiveness, adoption, implementation and maintenance (Glasgow et al., 1999) of Entre Nosotras, which are described below (Box 1; Figure 1b).

## Participants and procedures

Participants in the process evaluation were sampled from the 225 women who participated in the feasibility trial. Adult women residing in the study community with distress levels classified as moderate or below using the Kessler-6 assessment (Kessler-6 score <13) were eligible to participate in the parent study. An analysis of the psychometric properties of the Kessler-6 in this sample revealed that, with modifications, the Kessler-6 displayed adequate internal and external construct validity. However, the Kessler-6 had low internal consistency, which was largely due to the low item-rest correlation of the fifth item ('feeling like everything was an effort') (Greene et al., 2023).

Study participants were identified and recruited by HIAS staff along with their network of community leaders and partners. In the Ecuador sites, refugees, migrants and members of the host community were included. In Panamá, refugees and migrants were included. All participants were identified and recruited by HIAS staff. We selected up to 10 participants per study community to participate in qualitative interviews after completing the intervention. Participants were selected using maximum variation sampling to reflect a range of perspectives. We stratified all participants by the following categories and randomly selected participants within each strata using a random number generator: (1) high vs. low levels of distress at baseline; (2) high vs. low intervention attendance; (3) refugee or migrant vs. host community member (in Ecuador) and (4) study community (i.e., each of the 11 study communities). All intervention facilitators were invited to complete a qualitative interview after they completed their intervention groups.

Semi-structured interviews were designed to capture information about the following implementation outcomes: acceptability, relevance/appropriateness, feasibility, reach and accessibility, effectiveness, safety, implementation (including barriers and facilitators) and sustainability. Questions were informed by the Johns Hopkins Dissemination and Implementation Science Measure (Haroz and Murray, 2018; Aldridge et al., 2022). Interviews lasted approximately 45 min and were conducted by a member of the research team in Spanish either in person (48.6%) or by phone (51.4%). To complement the findings from qualitative interviews, the research team used the FRAME-IS to capture adaptations made across phases of implementation (Stirman et al., 2019; Miller et al., 2021).

All procedures were approved by the Institutional Review Boards at Columbia University Irving Medical Center (United States), Universidad de Santander (Panamá) and Universidad San Francisco de Quito (Ecuador). The trial protocol was published and registered online (NCT05130944) (Greene et al., 2022a).

## Analysis

All qualitative interviews were coded by three researchers who were fluent in Spanish. Initially, the coders reviewed 12 transcripts (2 facilitators per site, 2 participants per site) to develop the codebook. Twenty-six themes and 85 subcodes were generated using a thematic analysis approach and they were then mapped onto the

**Table 1.** Implementation strategy specification

| Definition | ERIC match | Actor | Action | Target | Temporality | Dose | Outcomes affected | Justification |
|---|---|---|---|---|---|---|---|---|
| Engage community stakeholders | Build a coalition; inform local opinion leaders; Involve executive boards; conduct local consensus discussions | IO | Develop relationships with stakeholders in each study site; collect data on local priorities and appropriate intervention strategies; co-design intervention; present the implementation plan to community leaders and boards; Consult with the community regarding implementation decisions throughout implementation; Disseminate and discuss findings and experience with the program | COM | PRE, IMP, POST | Continuous | EFF, ADO, IMP, MNT | Engaging community stakeholders in co-designing the intervention and its implementation will increase ownership and optimize the intervention to each implementation context |
| Iterative co-design of the intervention | Tailor strategies; conduct small cyclical tests of change; promote adaptability; develop educational materials | IO, COM | Conduct iterative co-design workshops to develop and tailor the intervention and materials to each implementation context; Include guidance for adaptation within intervention manual; Include guidance for adapting to different delivery formats (e.g., remote, hybrid vs. in-person delivery) | FAC | PRE, IMP | Weekly during PRE | EFF, ADO, IMP, MNT | Co-designing the intervention with stakeholder to fit each context and community will promote intervention usability and fit. Iteratively adapting the intervention prior to implementation will enhance tailoring. |
| Community-based and group delivery format | Change service sites; change physical structure and equipment | IO, FAC | Identify locations within study contexts that are accessible to participants and provide a safe location for women to meet in groups | PAR | PRE | Once[a] | REA, IMP, MNT | Delivering the intervention in community settings and in groups will increase accessibility and leverages community resources to support sustainability |
| Task sharing with ongoing training and supervision | Create a new clinical team; develop and distribute educational materials; make training dynamic; Provide clinical supervision | IO | Identify women in the community who are motivated to work as intervention facilitators and are respected by other members of the community; Train selected women in the intervention and its implementation through an initial 2-week training; Provide ongoing, regular supervision and support to facilitators[b] | FAC | IMP | Once, Weekly[c] | ADO, IMP, MNT | Community members with adequate training and support will deliver the intervention with fidelity and promote its acceptability, appropriateness and sustainability |
| Provide financial compensation | Use other payment schemes | IO | Provide stipends to facilitators to compensate them for their time in training, preparation and implementation of the intervention. Reimburse participant costs related to participation including transportation or internet connectivity/communication costs. Stipends and reimbursement rates aligned with policies established by the implementing organization | FAC, PAR | IMP | Weekly | REA, ADO, IMP | Compensating facilitators for their time and reimbursing participant costs will reduce barriers to participation |

Abbreviations:

- Actor/Target: IO – staff at the implementing organization (NGO) who managed the program; FAC – intervention facilitators; PAR – study participants; COM – community
- Temporality: PRE – pre-implementation; IMP – implementation; POST – post-implementation
- Outcomes: REA – reach; EFF – effectiveness; ADO – adoption, IMP – implementation, MNT – maintenance.

[a]Some sites required that we identify new locations for the intervention during implementation due to safety and security concerns. In Guayaquil (urban setting) and Tulcán (rural setting), sessions were held in a community house. In Panamá (urban and peri-urban setting), sessions were held in the HIAS Office.
[b]In Ecuador, all facilitators were members of the community where the study was being implemented. In Panamá, facilitators pairs included one member of the community and one person with some training in mental health.
[c]The 2-week training of facilitators was provided once and was followed by weekly supervision sessions.

1a.

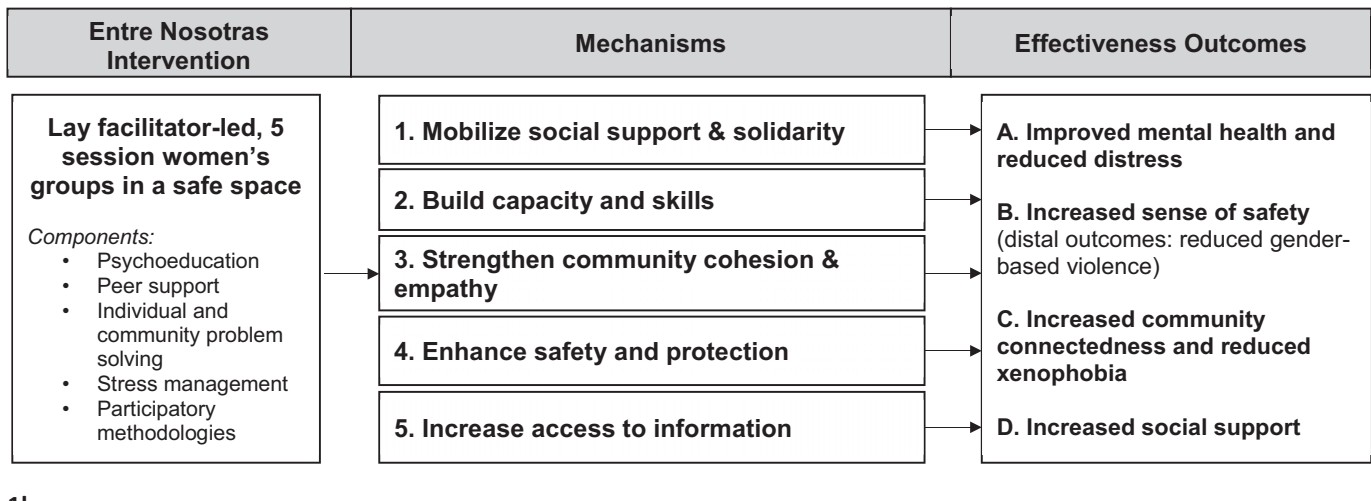

1b.

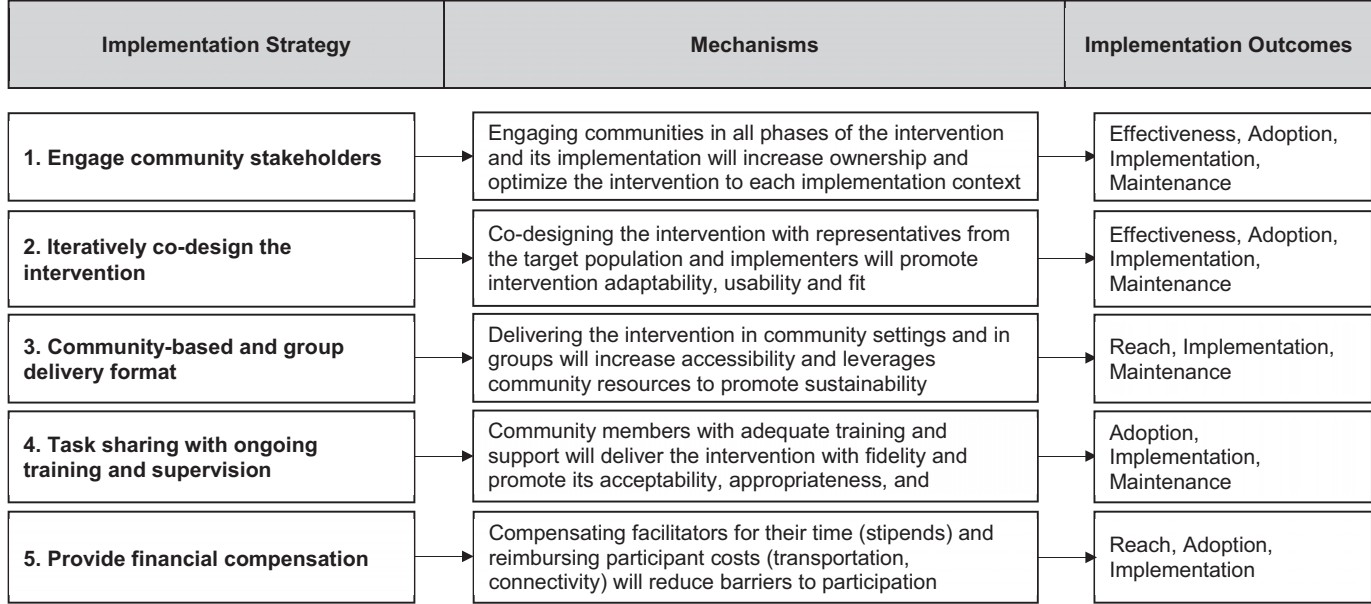

**Figure 1.** Intervention (a) and implementation (b) theories of change.

enhanced PRISM framework, which includes domains covering key implementation determinants as well as the RE-AIM implementation outcomes (Glasgow et al., 1999; McCreight et al., 2019). After the codebook was developed, three researchers independently pilot coded a subset of the study transcripts to evaluate intercoder reliability. We continued to pilot code transcripts until the coders achieved 98.4% agreement, which required coding 10% of all transcripts. After achieving sufficient agreement, the three researchers distributed the remaining transcripts and applied study codes. After all transcripts were coded, the researchers generated memos with a summary of the findings within each theme and subcode along with illustrative quotes. The research team met weekly to discuss the recommended changes to the codebook and the memos. All coding was done using NVivo. We explored the alignment of the qualitative themes with the intervention and implementation theories of change (Figure 1a,b; Table 2) to explore whether the intervention components and implementation strategies contributed to implementation outcomes as hypothesized.

We described each adaptation made to the intervention and/or its implementation and the goal/reason the adaptation was made using the domains described in the FRAME-IS tool (Miller et al., 2021). We coded each adaptation according to where the adaptation was made (Guayaquil, Panamá, Tulcán); when the adaptation was made (pre-implementation, during implementation); who made the decision to make the adaptation (community, practitioners, researchers); what type of adaptation was made (contextual, intervention content, training and evaluation, implementation) and what contextual factors contributed to the need for adaptation (sociopolitical, organizational/setting, provider, recipient factors).

## Results

Seventy-seven participants and thirty facilitators participated in the process evaluation interviews. Participants were 35.2 years (SD = 12.0) years of age, on average. Participants were from

Venezuela (75.0%), Ecuador (11.8%), Colombia (9.2%) and Central American or Caribbean countries (4.0%). The majority had migrated for economic reasons (59.1%), followed by family reasons (21.2%), violence or armed conflict (7.6%) or other reasons (12.1%). Then, 40% of participants were living in Tulcán followed by Guayaquil (34.3%) and Panamá (25.7%). Most participants had a primary school education or higher (89.5%) and were unemployed (63.2%). Process evaluation participant demographics (age, education, employment, country of origin and current location) as well as baseline levels of psychological distress were comparable to the full feasibility trial sample ($p > 0.05$) (Greene et al., 2023). Forty percent of the 30 facilitators were from Tulcán (40.0%; n = 12), followed by Guayaquil (33.3%; n = 10) and Panamá (26.7%; n = 8). Participants described a range of factors that contributed to reach, effectiveness, adoption, implementation and maintenance of the intervention. Below, we present the aspects of the study context and implementation of the intervention that promoted or hindered these implementation outcomes (Figure 1a,b).

### Reach and retention

Reach and retention describe the number and representativeness of women who participated (i.e., reach) and remained engaged (i.e., retention) in the Entre Nosotras interventions and the factors that influence participation (Holtrop et al., 2021). Most participants who participated in the process evaluation interview attended at least one session (85.7%) and more than half completed the intervention (59.7%; i.e., attended four to five sessions). The mean and median number of sessions attended were 3.2 (SD = 1.9) and 4.0 (interquartile range: 1,5, respectively. Participants described factors related to the intervention setting, social context and delivery format that influenced program reach and retention (Table 2). Several participants described that community insecurity was an important barrier to participating in the intervention, particularly in Guayaquil. Engaging community and organizational stakeholders to determine safe and accessible intervention locations within communities was critical to increasing program reach and retention.

> *"There is so much crime right now. Imagine that when you leave your house they kill. Things happen. They kill us when you put one foot outside or even sometimes inside. You always have to be prepared."*
> – Intervention participant, host community, age 59, Guayaquil

Adapting delivery settings and formats to the changing context (e.g., transitioning to remote delivery during COVID-19 lockdowns) influenced initial and continued access to the intervention. The impact of these strategies was not experienced equally. For example, while remote delivery increased access for some participants, others who did not have stable access to internet or technology were less likely to engage or consistently participate with the program.

> *"I thought [the remote format] was good because of the pandemic. Well, you have to take care of yourself, but it went well. I think I was the one who had the least connectivity because of the internet and it cost me a lot, well it cost me a little. It was a little difficult, but praise God that I managed to connect and that I managed to participate in part, not in all, but in a large part of the sessions we had, and it was excellent. I really loved it."*
> – Intervention participant, migrant or refugee, age 37, Guayaquil

Some safe community settings where intervention sessions were held were not accessible for people with disabilities despite being located close to participants' homes. Other subgroups had limited access to the intervention, and these underrepresented subgroups varied across sites. For example, by design, host community

members were not included in Panamá. This was primarily due to the relationships that existed between HIAS, the implementing organization and community. The HIAS Ecuador office had a long history of established partnerships with host community leaders, whereas in Panamá, which was a newer site for the HIAS, these relationships were in the process of being formed and their programs primarily focused on refugee and migrant communities at the time the study was conducted. Across the sites in Ecuador, we observed variation in the proportion of the sample that represented the host community such that participants in Tulcán were more likely to be members of the host community (30.0%) than in Guayaquil (9.9%). Community engagement and outreach by participants and the implementing organization were considered key strategies for expanding reach and engaging underrepresented subgroups in the intervention.

### Effectiveness

Effectiveness is the self-reported impact of the Entre Nosotras intervention on individual psychosocial, protection and other unintended outcomes as well as variability in these impacts across subgroups (Holtrop et al., 2021). Participants described four types of intervention mechanisms that contributed to improvements in psychosocial and protection outcomes and aligned with the intervention theory of change (Figure 1a). These mechanisms included psychological processes (e.g., building the capacity and skills to cope with adversity), social mechanisms (e.g., mobilizing social support and community connectedness/empathy), protection mechanisms (e.g., enhancing safety and protection) and functional mechanisms (e.g., increasing knowledge about local resources, participant rights and available services). Most of the intervention components, including stress management, relaxation/breathing techniques, problem solving, psychoeducation and learning ways to support each other, were perceived to activate the psychological mechanisms of change and lead to improved mental health, social support and community connectedness. Intervention activities dedicated to supporting others in the community and strengthening support networks activated the social mechanisms, as hypothesized, and were perceived to result in improved social support, community connectedness and mental health. Participants reported that problem solving and developing protection pathways (i.e., safety plans/maps) led to increased sense of safety and social support. Finally, participants described that psychoeducation and specifying protection pathways and safety plans increased their knowledge of what resources were available in the community and thus improved their mental health and sense of safety.

> *"What I liked most was the map of security … If you have a problem you have to go look for [a solution]. Many times we do not know where to go. We stay quiet and calm. But in the map we made, we see where we can go and how we can solve the problem better."*
> – Intervention participant, host community member, age 45, Tulcán

Beyond the intervention components, some aspects of the implementation strategies also influenced the perceived effectiveness of the intervention. Participants noted the importance of the group delivery format. They reported gaining motivation and a stronger support system by participating in the intervention as a group. One participant also noted that having mixed groups including migrant, refugee and host community members has the potential to build empathy and reduce xenophobia.

> *"Well, one of the impacts of this [intervention], which I had never achieved, is that I feel that here I have made friendships that I didn't*

**Table 2.** Relationships among implementation strategies and outcomes

| Strategy 1. Engage local community members and organizational stakeholders | | |
| --- | --- | --- |
| PRISM domain | Outcomes | Description |
| External context | Reach/retention | (+) Women were more likely to participate if the community and session locations were safe and protected women |
| | Maintenance | (+) Strengthened sense of agency and encouraged community participation and ownership |
| Internal context (perspectives) | Maintenance | (±) Maintenance of intervention is subject to group dynamics, priorities and continued engagement |
| Internal context (implementation) | Reach/retention | (−) Underutilized community and social networks to advertise program |
| | Adoption | (+) NGO role was valuable for facilitators and supported implementation |
| | Implementation | (+) NGO provided critical resources, logistical and community outreach support |
| Strategy 2. Iterative co-design the intervention to promote adaptability, usability and fit | | |
| 2a. Conduct a series of human-centered design workshops to iteratively develop, refine and pilot the intervention with members of the community. | | |
| 2b. Design intervention with guidelines and flexibility for adapting to each implementation context and different modalities (e.g., virtual, in person, hybrid) | | |
| PRISM domain | Outcomes | Description |
| External context | Reach/retention | (±) Session flexibility and adaptability made it more feasible to attend sessions, but issues related to limited time, childcare and other responsibilities could not be fully overcome |
| | Maintenance | (+) Strengthened sense of agency and encouraged community participation and ownership |
| Internal context (characteristics and perspectives) | Reach/retention | (±) Remote/hybrid delivery increased access for some who were not able to attend in person due to other responsibilities or transportation issues, but decreased access for people without a stable internet connection or technology that would enable them to join remotely [Also related to PRISM Domain: Overarching Issues] |
| | Implementation | (+) Flexibility to work (facilitators) and participate (participants) in session from home was helpful |
| Internal context (implementation) | Effectiveness | (−) In-person sessions offered better potential to connect with others, motivation and make referrals to additional resources. Facilitators noted the importance of reading body language to accomplishing these objectives, which was difficult when sessions were conducted remotely [Also related to PRISM Domain: Fit of intervention] |
| | Implementation | (+) Adaptability of the intervention implementation enabled facilitators to tailor aspects of program to make content more accessible |
| | | (+) Complying with COVID-19 regulations alleviated concerns about risks of attending sessions |
| | Maintenance | (+) Manual encourages participants to tailor intervention to local needs for continued implementation |
| Strategy 3. Community-based and group delivery format | | |
| PRISM domain | Outcomes | Description |
| Internal context (characteristics and perspectives) | Reach/retention | (±) Having the intervention sessions take place in the communities and close to some women's home increased access for some. However, for others (e.g., people with disabilities) some of the locations were inaccessible |
| | | (+) Group format met participant expectations and encouraged them to participate and continue to attend sessions |
| | Effectiveness | (+) Group discussions were motivating, helped them develop a better support system, and was seen as a critical factor to improving their mental health. |
| | Implementation | (−) Some participants were not comfortable sharing their problems with others in a group setting |
| | | (+) Attending the sessions outside their home and within communities helped them disconnect from their other worries/responsibilities and focus on the information in the sessions |
| Internal context (implementation) | Reach/retention | (+) Participants promoted or were eager to promote the intervention within their communities to expand reach |
| | | (+) Implementing organization outreach and provision of resources was valuable for increasing reach, recruiting a broader demographic and support general implementation |
| | Implementation | (+) Having a dedicated and reliable space with necessary materials within the community was critical for implementation |
| | | (+) The community-based nature of the intervention enabled facilitators to form positive relationships with participants characterized by good communication and empathy |
| Strategy 4. Task-sharing (training of nonspecialists to deliver the intervention) with ongoing training and supervision | | |
| PRISM domain | Outcomes | Description |
| | Adoption | (+) Receiving training, materials and ongoing support by supervisors/program staff enabled the facilitators to deliver the intervention as intended |

*(Continued)*

**Table 2.** (*Continued*)

| Strategy 1. Engage local community members and organizational stakeholders | | |
|---|---|---|
| PRISM domain | Outcomes | Description |
| Internal context (characteristics and perspectives) | Implementation | (+) Facilitator characteristics were critical to the successful implementation of the program: motivation, enthusiasm, responsible, competent, empathetic, dedicated and having enough time for their role |
| | | (+) Having facilitators from their community led to greater satisfaction among participants and these facilitators served as important linkages to other services/supports through the implementing organization and other resources |
| | Maintenance | (+) Having facilitators and participants come from the same community may encourage them to tailor implementation to improve the program |
| Internal context (implementation) | Implementation | (−) Facilitators faced challenges to implementation, some of which they felt could have been addressed with further training: poor communication or disagreements between co-facilitators, challenges involving participants (e.g., lack of motivation, lateness), holding online sessions, difficulties with the content and not knowing how to manage complex situations with participants (e.g., emergencies) |
| Strategy 5. Provide financial compensation | | |
| 5a. Provide stipends to facilitators | | |
| 5b. Reimburse participants for transportation and connectivity/communication costs | | |
| PRISM domain | Outcomes | Description |
| Internal context (implementation) | Reach/retention | (−) Additional economic support would encourage attendance and participation |
| | Effectiveness | (+) Economic support/reimbursement for participants had a positive impact on the perceived impact and accessibility of the intervention |
| | Adoption | (+) Compensating facilitators enabled them to deliver the intervention and was helpful given that many were unemployed |
| | Implementation | (±) While the reimbursements were appreciated, additional economic support and livelihood/skills training components for participants would improve the relevance of the program to help meet their basic needs |

Abbreviations: PRE, Pre-implementation; IMP, Implementation; POST, Post-implementation; +, Promoted implementation outcome; −, Hindered implementation outcome.

*have before. And I know that I can always count on them for anything, any circumstance."*
  – Intervention participant, migrant or refugee, age 32, Panamá

*"They [the host community] have a lot of xenophobia. People flee us here and many do not treat us. Many treat us like we are thugs, that we are bad people, that women are whores. Maybe having more Ecuadorian women [in the groups] would be good, and we would be able to raise awareness and talk about xenophobia."*
  – Intervention participant, migrant or refugee, age 26, Tulcán

Having the group sessions take place in-person provided opportunities for participants to connect with each other and leverage their collective networks and resources. Facilitators noted that in-person sessions enabled them to deliver the intervention more effectively because they could read participant's body language and connect with group members more effectively than they could online. Participants and facilitators referenced the economic support and/or reimbursement they received as having a positive impact while also making the intervention more accessible.

### Adoption

Adoption is the degree to which the facilitators use and implement the intervention as well as their reasons for doing so (Proctor et al., 2011; Holtrop et al., 2021). Aspects of the implementation strategies that influenced facilitator adoption of the program were closely related to the relationship and interaction among the implementing organization, the facilitators and the community. The support provided by the implementing organization for facilitators encouraged their adoption of the program and their motivation to deliver

the intervention. The implementing organization had preexisting relationships with the study communities and played a central role in engaging and identifying community members to be trained as facilitators, following a task-sharing model. The facilitators explained that the training, materials and ongoing support they received by supervisors and program staff enabled them to deliver the intervention as intended. Additionally, the compensation the facilitators received through a stipend provided by the implementing organization enabled them to dedicate time to preparing for and delivering the intervention.

*"Look, for me it was a unique experience because I have always been a closed person. I don't say or express my things. And the truth is that since I found out about the program, I didn't hesitate to tell them I was interested. Why? Because that phrase 'Entre Nosotras' caught my attention. This was something like that - between women, between us – that phrase came to me and told me like now you have access to express yourself. Because I don't have any relatives here, at least not my mother and my sister who are closest to me, the support from [the implementing organization] and the girls in the group really felt like I was with my family and it was a truly unique experience. Sometimes people ask me what it is like, 'Entre Nosotras', and it is really something that, just as I faced my fears, I faced everything that trapped me."*
  – Facilitator, Guayaquil

### Implementation

Implementation is the fidelity to different elements of the intervention's functions or components, including consistency of delivery as intended, the time and cost of implementation and adaptations to the intervention and implementation strategies (Holtrop et al.,

2021). All five strategies were perceived to influence the implementation of the intervention. Engaging community and organizational stakeholders in community outreach, logistical support and provision of resources was perceived as an important contributor to the successful implementation of the intervention. Moreover, participants reported that having members of their community deliver the intervention through the task-sharing model improved its implementation and participant satisfaction because they were able to identify with each other's lived experience and found it empowering to see members of their community in the position of a facilitator.

> *"It's really great that the girls in the community themselves are supporting these other women who often don't know about these programs that really help to motivate them to get ahead."*
> – Intervention participant, migrant or refugee, age 49, Guayaquil

The intervention's adaptability and community-based format helped to overcome barriers to participation, engagement and delivery and to tailor the intervention and its implementation to the local context. Most participants reported that these strategies benefited implementation and allowed them to find trust in a group that included other migrant and refugee women, including from their country of origin and members of the host community. However, one participant stopped attending after one session and described that she was not comfortable sharing her problems with others in a group setting because there were concerns about potential violations of privacy, particularly in groups with migrant, refugee and host community women.

> *"The women - Ecuadorian, Colombian, Venezuelan women - give us support. They help us, lend us a hand, and they do not despise us. Everything that is talked about in the group stays there, it does not go elsewhere, because you created that group for us to unburden ourselves of things that have happened with them and with us… So let it be like that, let everything stay there, among us ('Entre Nosotras')"*
> – Intervention participant, migrant or refugee, age 18, Tulcán

We recorded 10 adaptations to the intervention (Table 3). Five of these adaptations occurred across all intervention sites. These adaptations included adapting intervention materials for remote delivery due to the COVID-19 pandemic, modifying the scope of the intervention to fit community priorities (e.g., adding in activities focused on community safety and protection), changing the intervention format from the originally planned individual intervention to group-based intervention, providing mechanisms for childcare to facilitate participation, and disseminating intervention materials and exercises digitally using WhatsApp to supplement paper-based versions. Two adaptations varied across country contexts. In Ecuador, the target population was expanded from focusing solely on migrants and refugees (original plan) to also including host communities. Similarly, in Ecuador, the communities that were included in the study had to be redefined due to mobility among the migrant and refugee populations to ensure the intervention remained accessible. Three adaptations were specific to a single context. In Guayaquil, several aspects of the implementation of the intervention required adaptation due to security concerns as well as participant availability and competing priorities (e.g., work schedules). In Tulcán, additional adaptations to the training materials and additional training time were required due to more variable literacy levels among participants and facilitators.

> *"In my training group there were some [facilitators] who were illiterate, they could not read or write. So it was a challenge… We worked with the women who didn't know how to read. And [the trainers] explained everything clearly and kindly, so they know how to do it. They also fulfilled all the activities."*
> – Facilitator, Tulcán

Six of the ten adaptations were made by pre-implementation. All pre-implementation adaptations related to sociopolitical factors, including societal/cultural norms and preferences, existing policies, the political climate and social context and the COVID-19 pandemic. The pre-implementation adaptations were determined by a range of stakeholders (community, practitioners, researchers) and modified elements of the study context, intervention content, training, implementation and evaluation. Four of the 10 adaptations were made during implementation. All these adaptations were recommended by practitioners and/or community members and targeted implementation processes or aspects of the training and evaluation procedures. Adaptations made during implementation were driven by organization/setting, provider or recipient factors.

### Maintenance

Maintenance, or sustainability, is the extent to which the implementing organization and facilitators incorporate the Entre Nosotras intervention into routine programming (Proctor et al., 2011; Holtrop et al., 2021). In general, stakeholder engagement, iterative co-design of the intervention and task sharing strategies generated ownership and motivation among participants and facilitators to sustain the intervention within their communities. Facilitators explained that these strategies enabled them to modify aspects of the intervention content and implementation to fit the dynamic community needs and context. However, participants and facilitators recognized that the resources and processes that the intervention required meant that maintenance of the intervention would be subject to group dynamics, priorities and continued engagement. Participants and facilitators identified resources and actions that both the implementing organization and the community would need to commit to ensure sustainment of the intervention.

> *"To implement a program like this in the community requires resources, which I do not have. Of course, then it would be through institutions that can organize this and I would be willing to contribute my knowledge, my preparation, and everything I have experienced in this program."*
> – Facilitator, Panamá

> *"I don't know if [the implementing organization] has contact or if they can talk to someone from the community boards, which is closest to the community, to see if they have an opening toward these sessions or if they can spread the word in the community so that people have more knowledge and want to participate."*
> – Intervention participant, migrant or refugee, age 24, Panamá'

Despite the additional resources required to continue implementation of the intervention, both participants and facilitators described how their group had maintained their support for each other and continued to meet.

> *"We meet like this, with the children in the park for them to play and for us to talk. And by WhatsApp, we send each other messages like: Good morning, how are you? How are you feeling? Is there anything we can help you with?"*
> – Facilitator, Tulcán

### Discussion

This study described a qualitative examination of strategies used to promote the reach and retention, effectiveness, adoption, implementation and maintenance of a group psychosocial intervention for migrant, refugee and host community women in Ecuador and Panamá. The strategies were designed to address anticipated barriers and facilitators within each of the study contexts and did not

**Table 3.** Summary of adaptations to the intervention and its implementation

| Description | Where<br><br>GYE: Guayaquil<br>PAN: Panama<br>TUL: Tulcan | When<br><br>PRE: Pre-implementation<br>IMP: Implementation | Who<br><br>COM: Community<br>PRAC: Practitioner(s)<br>RES: Researcher(s) | What<br>CON: Contextual<br>INT: Intervention content<br>TE: Training and Evaluation<br>IMP: Implementation | Goal/reason | Contextual factors<br><br>SOC: Sociopolitical<br>ORG: Organizational/setting<br>PROV: Provider<br>REC: Recipient |
|---|---|---|---|---|---|---|
| Changed eligibility to include migrant, refugee and host community members | GYE, TUL | PRE | RES, PRAC, COM | CON | Increase reach, engagement and community integration | SOC: Societal/cultural norms and values; REC: Immigration status |
| Adapt some training, recruitment, data collection, intervention and community engagement activities for remote delivery | GYE, PAN, TUL | PRE | RES, PRAC | TE, IMP | Social distancing guidelines and promote safety. In GYE, all activities transitioned to remote implementation in January 2022 | SOC: Existing policies, COVID-19 |
| Identify community leaders, organizational staff and others to accompany research team for interviews | GYE | PRE | RES, PRAC | IMP | Ensure safety of research team and participants during study activities | SOC: Political climate and social context |
| Broaden the protection component of the intervention to not focus solely on intimate partner violence | GYE, PAN, TUL | PRE | RES, PRAC | INT | Improve fit of the intervention with community priorities identified during formative research | SOC: Societal/cultural norms and values |
| Change intervention format from individual to group-based intervention | GYE, PAN, TUL | PRE | RES, PRAC | INT, IMP | Align format of the intervention to community preferences identified during formative research | SOC: Societal/cultural norms and values |
| Resampled communities to ensure access for migrant and refugee populations | GYE, TUL | PRE | RES | CON | Increase reach and engagement; population mobility reduced the number of migrants and refugees in some communities due increased rental prices, migration and so forth | SOC: Political climate and social context; ORG: Location accessibility, REC: Crisis or emergent circumstances |
| Modify training materials to be more interactive and accessible for people with varying levels of literacy | TUL | IMP | PRAC | TE | Increase engagement and competencies by making the training materials and activities more accessible | PROV: Previous training and skills; cultural norms, competency |
| Establish a mechanism for childcare during sessions, including partnering with nearby child friendly spaces or having a co-facilitator care for children | GYE, PAN, TUL | IMP | PRAC, COM | IMP | Increase engagement during sessions for women interested in attending with young children | REC: Access to resources |
| Disseminate intervention materials and exercises through WhatsApp (vs. only paper-based versions) | GYE, PAN, TUL | IMP | PRAC | IMP | Participants requested that the facilitators provide intervention information over WhatsApp | REC: Motivation and readiness |
| Facilitators delivered intervention sessions to evening and weekends to accommodate participant schedules | GYE | IMP | PRAC | IMP | Increase reach or engagement to overcome barriers to attendance (e.g., work, childcare) | ORG: Service structure, time constraints |

use an existing framework for specifying and selecting implementation strategies. However, the strategies we selected aligned closely with many of the Expert Recommendations for Implementing Change (ERIC) strategies (Powell et al., 2015; Waltz et al., 2015) and were similar to strategies that have been used in previous mental health or humanitarian health implementation research (Cohen and Yaeger, 2021; Wood and Kallestrup, 2021).

In general, these strategies were perceived by intervention facilitators and participants to lead to improved reach and retention, effectiveness, adoption, implementation and/or maintenance of the Entre Nosotras intervention. We found that these strategies seemed to improve implementation outcomes by modifying processes within the inner context, including participant and facilitator characteristics, preferences and behaviors as well as aspects of the intervention delivery (McCreight et al., 2019). Factors and processes within the external context that influenced the implementation of Entre Nosotras included community safety and security, competing priorities and responsibilities and community engagement and ownership. Only two strategies interacted with aspects of the external context (e.g., the external environment) to modify implementation outcomes (McCreight et al., 2019). By engaging community stakeholders throughout the implementation phases, participants reported feeling safer attending sessions in the community, thus promoting intervention reach. Similarly, the continued engagement fostered a sense of agency and ownership over the intervention, which was perceived to increase the potential for intervention maintenance. The second strategy that was perceived to interact with determinants in the external context was the iterative co-design of the intervention to promote adaptability, usability and fit. This process, which is described in further detail elsewhere, involved an extensive formative research process to align the intervention with community priorities and series of iterative co-design workshops and mock sessions with community members, many of whom were later trained to be intervention facilitators (Greene et al., 2022b). While the adaptability of the intervention did enable facilitators to work around complex barriers to intervention delivery, it was not possible to fully overcome all external context barriers by having a highly adaptable and tailored intervention.

Factors and processes within the internal context that were critical to implementation included the modality (e.g., virtual or in-person), location and personnel involved in the implementation of the intervention. Multiple strategies were designed to modify these factors, including the group and community-based format, having facilitators be from the same community as participants, and designing flexible session activities that can be implemented in a virtual or hybrid format. The role of the implementing organization emerged as a central determinant of adoption, implementation and maintenance of the intervention. Specifically, the facilitators described the importance of the ongoing technical and operational support they received from the implementing organization. When discussing maintenance of the program, the implementing organization was seen as critical to the ability to sustain the implementation of Entre Nosotras. Furthermore, providing financial compensation helped to address many of the barriers to implementation and participation that existed within the external and internal contexts.

Notably, several implementation strategies also appeared to interact with elements of the intervention theory of change. For example, strategies related to stakeholder engagement, intervention co-design and task sharing may have directly promoted the social mechanisms of the intervention to further activate social networks and resources to promote mental health and psychosocial outcomes. Further examination of the interaction between intervention mechanisms and implementation mechanisms of change is needed to understand how the elements of an intervention itself may be modified by the implementation strategies and processes.

Most of the documented adaptations to the intervention were made by pre-implementation. It is possible that adaptations were made by facilitators during implementation that were not recorded and thus not captured in this study. All documented adaptations made to the intervention itself were consistent across sites and were made pre-implementation. As part of the stakeholder engagement and iterative intervention co-design strategies, the research team carried out a series of mock sessions to iteratively refine and optimize the intervention content prior to implementation (Greene et al., 2022b), which may explain why there were no documented adaptations made during implementation. Most of the adaptations that differed across sites were related to contextual or implementation adaptations. The insecurity and COVID-19 pandemic were particularly challenging in Guayaquil during the implementation phase, which prompted several implementation adaptations, including distributing intervention materials and conducting sessions virtually and having staff members provide security support during study activities. Differences in the characteristics of the population in Tulcán as compared to our other two study sites, including lower literacy and education levels, likely contributed to the need for adaptations to the training and intervention materials to make them more usable by facilitators and participants. These similarities and differences between the study sites highlight the unique role of context, particularly in the implementation of mental health and psychosocial interventions in diverse communities.

Several of our study findings align with previous implementation research examining mental health and psychosocial interventions in humanitarian contexts. A recent review identified that most prior research employing task-sharing models in humanitarian settings have trained members of the displaced community as intervention facilitators. Few studies have also included host community members as lay providers (Cohen and Yaeger, 2021). Many of these studies have identified that the peer- and group-based nature of mental health and psychosocial interventions is a valuable aspect of the intervention format for study participants (Dickson and Bangpan, 2018; Cohen and Yaeger, 2021; Harker Roa et al., 2023). Prior research has also reinforced the centrality of community stakeholder engagement to the successful implementation of mental health and psychosocial support in low-income and humanitarian contexts (Dickson and Bangpan, 2018; Greene et al., 2021; Harker Roa et al., 2023). Similarly, several intervention mechanisms of change have also been identified in evaluations of other group-based psychosocial programs. For example, a process evaluation of Semillas de Apego, a psychosocial program for caregivers in Colombia, similarly found that the group format formed and strengthened social support networks. Participants also described that mindful breathing and other relaxation techniques were useful skills to manage stress and promote well-being (Harker Roa et al., 2023).

The consistency of our qualitative findings with previous literature and our intervention and implementation theory of change models provides some assurance that the implementation strategies integrated into this study functioned as intended. However, this study has limitations that must be considered when interpreting these results. These results rely on qualitative interviews and lack a comparator, thus limiting our ability to isolate the effect of the

intervention on these implementation outcomes. We aimed to purposively select participants using a maximum variation sampling approach to ensure that we gathered the perspectives of people who completed the intervention as well as those who did not attend any or many sessions, those with variable levels of distress at baseline and to ensure representation across populations (host, migrant and refugee) and communities. Given these limitations, we are unable to determine the impact of individual strategies or a set of strategies on the implementation outcomes presented in this paper. These results are intended to generate preliminary findings regarding how participants and facilitators perceived elements of the intervention, its implementation and the broader, which context influenced diverse implementation outcomes.

## Conclusion

In this study, we explored the relationships among implementation determinants, strategies and outcomes as part of an evaluation of a community-based psychosocial intervention for migrant, refugee and host community women in three sites in Ecuador and Panamá. All implementation strategies and most adaptations were identified and incorporated into the implementation plan during the pre-implementation phase. Across the three diverse study sites, we found that aspects of the intervention itself remained relatively consistent. These findings suggest that it is feasible to scale-out interventions across settings with sufficient flexibility to adjust the implementation of these interventions to contextual realities. In this study, adjustments to the implementation and study context differed markedly across the study locations. Participants and facilitators perceived that community stakeholder engagement, iterative co-design of the intervention, delivering the intervention in community settings and a group format, training and supervising nonspecialist community members to deliver the intervention, and providing incentives to compensate for facilitator's time and reimbursing participants' expenses improved a range of implementation outcomes. These strategies may facilitate the scalability of the interventions by promoting their flexibility and adaptability to different settings and populations. Further research empirically evaluating and testing the impact of these implementation strategies on a range of implementation outcomes is needed to advance the evidence on how to optimally deliver community-based psychosocial interventions in humanitarian settings (Tol et al., 2023).

**Open peer review.** To view the open peer review materials for this article, please visit http://doi.org/10.1017/gmh.2024.29.

**Data availability statement.** The data that support the findings of this study are available from the corresponding author, MCG, upon reasonable request.

**Author contribution.** A.G.B., M.C.G. and W.A.T. contributed to conceptualizing the study. M.C.G., G.W., M.L., I.M.J., A. Angulo, A. Armijos, M.E.G., C.V., E.W.H., L.D., L.B., C.C., A.D.L.C., M.J.L., A. Moyano, A. Murcia, M.J.N., A.R., J.S., D.V., L.S.A., M.C., M.W., A.G.B. and W.A.T. contributed to data collection and/or analysis. M.C.G. drafted the initial manuscript. All authors provided critical revisions and approved the final manuscript for publication.

**Financial support.** This study was funded by the United States Agency for International Development (USAID) under the Health Evaluation and Applied Research Development (HEARD), Cooperative Agreement No., *AID-OAA-A-17-00002*. The trial sponsors had no role in data collection, management, analysis or interpretation. MCG was supported by a Career Development Award from the National Institute of Mental Health (K01MH129572).

**Competing interest.** The authors declare no competing interests.

**Ethics statement.** The trial was approved by the Institutional Review Boards at Columbia University Irving Medical Center (United States), Universidad de Santander (Panamá) and Universidad San Francisco de Quito (Ecuador). The trial protocol was published and registered online (NCT05130944) (Greene et al., 2022a).

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
