## [Editor Report]

Cambridge Prisms: Global Mental Health

GMH-23-0174

Peer Review Letter

“Strategies to improve the implementation and effectiveness of community-based psychosocial support interventions for displaced, migrant, and host community women in Latin America”

This article investigates the intricate relationship between implementation determinants, strategies, and outcomes of a community-based psychosocial intervention for displaced, migrant, and host community women in Ecuador and Panamá. By focusing on the unique implementation challenges in these contexts, the study contributes to important evidence to improve mental health and psychosocial support programs for displaced, migrant, and host community populations in all humanitarian settings, but specially in what is called now the “global south”. Also, this research agenda complements the growing literature evaluating the effectiveness of programs, interventions and policies that aim at improving the wellbeing and integration of families and persons displaced by violence poverty, and climate-related emergencies. More specifically, the study provides strategies to promote “implementation quality”, and offers implementation barriers and challenges frequently faced by community-based psychosocial interventions in contexts of extreme vulnerability. 

General comments:

1. On strategy #2, “promoting intervention adaptability, usability, and fit”: I suggest the authors revise this strategy. In my opinion, this strategy summarizes a “what”, and does not describe the “how”. In other words, it describes an outcome, but not an actual strategy to achieve an outcome.

2. Describing the training and supervision procedures or protocols would help the reader understand better some of the results discussing implementation strategies. In particular, for the reader would be valuable to have an analysis on how training and supervision procedures promoted (or not) implementation outcomes.

3. Also, I suggest the authors explain if there is a formal or explicit process, procedure or protocol that focuses on adapting the intervention to the context. If there is, it would be an important contribution to describe it. 

4. It would be a great contribution if, in the conclusions, the authors included some reflections on how the evidence they provide could also contribute to the question on how to scale-up and scale-out community-based psychosocial interventions.

Specific comments:

1. I suggest including the following recent studies that analyze barriers to health and access to health care services for migrants and refugees in a similar context: 

a. https://pubmed.ncbi.nlm.nih.gov/35675560/

b. https://journals.plos.org/plosone/article/file?id=10.1371/journal.pone.0282786&type=printable

2. Methods section:

a. Participants and procedures: 

i. I suggest describing the distribution of the 225 participants across available characteristics to compare the “population” with the analytical sample. This is important to show the internal validity.

ii. Mention the total N of facilitators.

iii. Describe the distribution across in person vs telephone interviews.

3. Results section:

a. I suggest the authors framing the description of attendance as an implementation outcome. Also, to describe the whole distribution of attendance (median, mean, standard deviation, P10, P90, etc.). If this is not possible, explain why. 

b. Page 6: Can you explain why host community were excluded in the Panamá implementation? Is this relevant? Does it speak to adaptability as an implementation strategy?

c. Page 7: About this: “Intervention activities dedicated to supporting others in the community and strengthening support networks activated the social mechanisms, as hypothesized, and were perceived to result in improved social support, community connectedness, and mental health.” Is this an implementation strategy? Could it be framed as so?

d. Page 7: Revise this definition: “Adoption is the degree to which the facilitators became involved in and delivered the program as well as their reasons for doing so (Holtrop et al., 2021).” This definition is confusing. 

e. Page 9: I find the definition of “maintenance” obscure. Is this the same as “sustainability” or “penetration” proposed by Proctor et al. (2011)? Available here: https://pubmed.ncbi.nlm.nih.gov/20957426/ 

4. Discussion section:

a. I suggest revising this article as part of the framing of the discussion: https://www.frontiersin.org/articles/10.3389/fpsyg.2023.1134094/full . In Particular, the following finding can be related to this study: 

i. “Several of our study findings align with previous implementation research examining mental health and psychosocial interventions in humanitarian contexts. A recent review identified that most prior research employing task sharing models in humanitarian settings have trained members of the displaced community as intervention facilitators. Few have also included host community members as lay providers (Cohen & Yaeger, 2021). Many of these studies have identified that the peer and group-based nature of mental health and psychosocial interventions is a valuable aspect of the intervention format for study participants (Cohen & Yaeger, 2021; Dickson & Bangpan, 2018). Prior research has also reinforced the centrality of community stakeholder engagement to the successful implementation of mental health and psychosocial support in low-income and humanitarian contexts (Dickson & Bangpan, 2018; Greene et al., 2021). 

b. It would be really valuable if the authors con explain in detail this: “In general, these strategies were perceived by intervention facilitators and participants to lead to improved implementation outcomes. We found that these strategies seemed to improve implementation outcomes by modifying processes within the inner context, including participant and facilitator characteristics, preferences, and behaviors as well as aspects of the intervention delivery (McCreight et al., 2019).”

c. Also, to summarize the main evidence or results around this: “The role of the implementing organization emerged as a central determinant of adoption, implementation, and maintenance of the intervention.”

d. It is very important that the authors explain or describe the design features that promotes adaptability: “The second strategy that was perceived to interact with determinants in the external context was designing the intervention to promote adaptability.”

e. Briefly mention the specific adaptations that are referred here: “Differences in the characteristics of the population in Tulcán as compared to our other two study sites likely contributed to the need for adaptations to the training and intervention materials to make them more usable by facilitators and participants.”